# Efficacy of Licensed Monoclonal Antibodies and Antiviral Agents against the SARS-CoV-2 Omicron Sublineages BA.1 and BA.2

**DOI:** 10.3390/v14071374

**Published:** 2022-06-23

**Authors:** Lia Fiaschi, Filippo Dragoni, Elisabetta Schiaroli, Annalisa Bergna, Barbara Rossetti, Federica Giammarino, Camilla Biba, Anna Gidari, Alessia Lai, Cesira Nencioni, Daniela Francisci, Maurizio Zazzi, Ilaria Vicenti

**Affiliations:** 1Department of Medical Biotechnologies, University of Siena, 53100 Siena, Italy; lia300790@gmail.com (L.F.); dragoni16@student.unisi.it (F.D.); federica.giammari@gmail.com (F.G.); camilla.biba@student.unisi.it (C.B.); maurizio.zazzi@unisi.it (M.Z.); 2Department of Medicine and Surgery, Clinic of Infectious Diseases, University of Perugia, 06129 Perugia, Italy; elisabetta.schiaroli@unipg.it (E.S.); annagidari91@gmail.com (A.G.); daniela.francisci@unipg.it (D.F.); 3Department of Biomedical and Clinical Sciences L. Sacco, University of Milan, 20157 Milan, Italy; annalisa.bergna@unimi.it (A.B.); alessia.lai@unimi.it (A.L.); 4Infectious Disease Department, USL SUDEST, Toscana, Misericordia Hospital, 58100 Grosseto, Italy; brossetti1982@gmail.com (B.R.); cesira.nencioni@uslsudest.toscana.it (C.N.)

**Keywords:** SARS-CoV-2, mAbs, nirmatrelvir, remdesivir, molnupiravir, microneutralization assay, cell-based assay, omicron sublineages

## Abstract

Newly emerging SARS-CoV-2 variants may escape monoclonal antibodies (mAbs) and antiviral drugs. By using live virus assays, we assessed the ex vivo inhibition of the B.1 wild-type (WT), delta and omicron BA.1 and BA.2 lineages by post-infusion sera from 40 individuals treated with bamlanivimab/etesevimab (BAM/ETE), casirivimab/imdevimab (CAS/IMD), and sotrovimab (SOT) as well as the activity of remdesivir, nirmatrelvir and molnupiravir. mAbs and drug activity were defined as the serum dilution (ID_50_) and drug concentration (IC_50_), respectively, showing 50% protection of virus-induced cytopathic effect. All pre-infusion sera were negative for SARS-CoV-2 neutralizing activity. BAM/ETE, CAS/IMD, and SOT showed activity against the WT (ID_50_ 6295 (4355–8075) for BAM/ETE; 18,214 (16,248–21,365) for CAS/IMD; and 456 (265–592) for SOT) and the delta (14,780 (ID_50_ 10,905–21,020) for BAM/ETE; 63,937 (47,211–79,971) for CAS/IMD; and 1103 (843–1334) for SOT). Notably, only SOT was active against BA.1 (ID_50_ 200 (37–233)), whereas BA.2 was neutralized by CAS/IMD (ID_50_ 174 (134–209) ID_50_) and SOT (ID_50_ 20 (9–31) ID_50_), but not by BAM/ETE. No significant inter-variant IC_50_ differences were observed for molnupiravir (1.5 ± 0.1/1.5 ± 0.7/1.0 ± 0.5/0.8 ± 0.01 μM for WT/delta/BA.1/BA.2, respectively), nirmatrelvir (0.05 ± 0.02/0.06 ± 0.01/0.04 ± 0.02/0.04 ± 0.01 μM) or remdesivir (0.08 ± 0.04/0.11 ± 0.08/0.05 ± 0.04/0.08 ± 0.01 μM). Continued evolution of SARS-CoV-2 requires updating the mAbs arsenal, although antivirals have so far remained unaffected.

## 1. Introduction

While worldwide vaccination has played a key role in the global control of COVID-19, both natural and vaccine-induced immunity have been shown to wane rapidly and be subject to escape by divergent virus variants [1,2]. In addition, a proportion of individuals could not be vaccinated due to specific underlying morbidity or personal choice. Thus, development of therapeutics for treatment and prevention of COVID-19 has been set as a public health priority, delivering at fast pace a number of monoclonal antibodies (mAbs) targeting the virus spike protein as well as three small molecule antivirals interfering with SARS-CoV-2 RNA synthesis (remdesivir, molnupiravir) or polyprotein cleavage (nirmatrelvir).

The viral enzymes have a high degree of conservation across SARS-CoV-2 lineages, thus antiviral activity is expected to be unaffected by viral variants, as preliminarily shown by limited in vitro data [3,4]. By contrast, the recent spread of the highly divergent SARS-CoV-2 omicron lineages has changed the landscape of mAbs activity with respect to previous virus variants. Namely, the BA.1 lineage lost susceptibility to the first developed bamlanivimab/etesevimab (BAM/ETE) and casirivimab/imdevimab (CAS/IMD) therapeutic mAbs combos as well as to the prophylactic cilgavimab/tixagevimab (CIL/TIX) combo while remaining partly sensitive to sotrovimab (SOT). However, SOT further decreased activity against the subsequent BA.2 variant which appears to have restored susceptibility to CIL and partly to IMD. Data about mAbs susceptibility for the minor BA.3 and the recently detected BA.4 and BA.5 lineages are scanty, with preliminary evidence for SOT activity against BA.3 [5] and partial or limited CIL activity against BA.4 and BA.5 [6,7]. According to the latest COVID-19 treatment guidelines (https://files.covid19treatmentguidelines.nih.gov/guidelines/covid19treatmentguidelines.pdf; updated 17 June 2022), the administration of BAM/ETE, CAS/IMD, and SOT mAbs is not recommended, due to the expected lack of activity against the dominating omicron lineages. Bebtelovimab, which retains activity against all SARS-CoV-2 variants, remains the only mAbs approved by the FDA and submitted for approval to the EMA for emergency use for treatment of COVID-19. In addition, CIL/TIX can be administered as pre-exposure prophylaxis. 

It must be noted that due to the pressure to deliver effective treatments, mAbs activity on contemporary SARS-CoV-2 lineages is inferred exclusively from in vitro data, while pivotal clinical trials were conducted during epidemic waves dominated by virus variants which have later disappeared. In addition, in vitro data have been generated by different methods including a variety of live virus or pseudovirus neutralization assays and surrogate tests such as SARS-CoV-2 spike binding measured by enzyme immunoassay or surface plasmon resonance. This has generated some data inconsistency across studies. In this study, we expanded our previous work [8] based on an ex vivo approach to test the licensed therapeutic mAbs BAM/ETE, CAS/IMD, and SOT against the omicron BA.1 and BA.2 lineages as well as against the ancestral B.1 strain and the previously dominating delta variant. By examining mAbs activity in post-infusion sera from treated patients in an authentic in vitro neutralization assay, we provide the best surrogate data for in vivo activity. Moreover, we tested on the same SARS-CoV-2 variants the three licensed antivirals, both with and without the P-glycoprotein (P-gp) inhibitor CP-100356, to further define their resilience to virus variability and current therapeutic potential. 

## 2. Materials and Methods

### 2.1. Patients and Sera

The study was approved by the local Ethics Committee and written informed consent was obtained from all the patients enrolled (Neutro-COVID observational study, protocol number 4069/21). The study was conducted in accordance with the Declaration of Helsinki. Patients undergoing mAbs treatment were enrolled consecutively and selected based on undetectable NtAb before therapy, independently from their vaccination status. A pair of patient sera was collected, one before (baseline to comply with the negative NtAb selection criterion) and another one-hour post mAbs infusion (to test mAbs activity against the different virus variants). Thirty sera from a previous study [8] were included with their original NtAb values, although a random selection of 15 sera were retested against the wild-type virus to ensure consistency across the two studies, yielding comparable results. 

### 2.2. Cells and Viral Stocks

VERO E6 (CRL 1586TM ATCC^®^, Gaithersburg, USA), an adherent cell line derived from African green monkey kidney, was used to propagate and titrate the viral stocks as previously described [9]. The same cell line was used in the live virus microneutralization and drug susceptibility assays. VERO E6 cells were propagated in DMEM High Glucose (Euroclone, Pero, Italy) supplemented with 10% Fetal Bovine Serum (FBS, Euroclone, Pero, Italy) and 1% of Streptomycin/Penicillin (PS) (Euroclone, Pero, Italy) in a humified incubator at 37 °C with 5% of CO_2_. The same medium, containing 1% of FBS instead of 10% (infection medium), was used in all viral infection experiments. Uninfected cell cultures were handled in a Biosafety Level 2 (BSL2) laboratory, whereas all the infection experiments were performed in a BSL3 containment. The SARS-CoV-2 B.1 (D614G) wild-type, delta and omicron BA.1 and BA.2 stocks, used to challenge the mAbs sera in neutralization experiments and to determine the antiviral activity of drug compounds in drug susceptibility assays, are detailed in Appendix A. Before performing the neutralization and phenotypic experiments, six replicates of 5-fold serial dilution of each viral stock were titrated in VERO E6 cells to determine the Tissue Culture Infectious Dose per milliliter (TCID_50_/_mL_), which is defined as the amount of virus required to infect 50% of replicate cell cultures as previously described [9]. The cytopathic effect and consequently the TCID_50_/_mL_ was determined by luminescence as described below. In addition, each viral stock was initially quantified by plaque assay as previously published [9].

### 2.3. Live Virus Microneutralization Assay

Microneutralization experiments were performed as previously described [8]. Briefly, after inactivation at 56 °C for 30 min, the patient serum was prediluted 1:5 and two-fold serial dilutions were prepared in 96-well format. One hundred 50% TCID_50_ of each SARS-CoV-2 viral stock were added to the sera and incubated at 37 °C with 5% CO_2_ for 1 h. Then, serum-virus mixtures were added to 5000 pre-seeded VERO E6 cells in 96-well plates and incubated at 37 °C with 5% CO_2_. After 72 h, the ability of sera to neutralize the virus was determined measuring the cell viability by the CellTiter-Glo 2.0 Luminescent Cell Viability Assay (Promega, Madison, WI, USA) with the GloMax^®^ Discover Multimode Microplate Reader (Promega, Madison, WI, USA) and the mAbs neutralization titer was expressed as the serum dilution corresponding to half-maximal inhibition of virus-induced cell death (ID_50_). Sera below 5 ID_50_ were scored as not neutralizing and given a 2.5 value for statistical analysis. 

Each serum was tested in technical duplicates in two independent experiments. Each plate included: (i) a mock infection control (uninfected cells); (ii) a virus control (infected cells without patient serum); (iii) a known SARS-CoV-2 neutralizing serum (positive control), yielding a median titer of 69 (59.3–69.9) in five independent runs. In addition, the virus test dose was confirmed by back titration, consisting of two-fold serial dilutions of each viral stock (100, 50, 25, 12.5, and 6.25 TCID_50_). The virus test dose was considered acceptable if the back-titration results were positive in at least 3 subsequent virus dilutions. For a run to be valid, the coefficient of variation for the technical duplicates and for the two independent experiments had to be both below 30%. The initial validation of the assay was performed with the First WHO International Standard [10] anti-SARS-CoV-2 immunoglobulin (Version 3.0, Dated 17 December 2020; code 20/268 NIBSC, Ridge, UK). 

### 2.4. Drug Susceptibility Assay

The P-gp inhibitor CP-100356 hydrochloride (MCE^®^ cat. HY-108347 distributed by DBA, Milan, Italy), Remdesivir (MCE^®^ cat. HY-104077 distributed by DBA, Milan, Italy), Nirmatrelvir (MCE^®^ cat. HY-138687 distributed by DBA, Milan, Italy) and EIDD-1931 (MCE^®^ cat. HY-125033 distributed by DBA, Milan, Italy), the active form of molnupiravir, were supplied as powder and dissolved in 100% dimethyl sulfoxide (DMSO). The VERO E6 cytotoxicity of CP-100356 hydrochloride alone and of the three antiviral drugs both with and without the addition of 0.5 µM CP-100356 was determined by the Cell Titer-Glo 2.0 Luminescent Cell Viability Assay (Promega) according to the manufacturer’s protocol. The luminescence values obtained from cells treated with the antiviral compounds or DMSO were measured through the GloMax^®^ Discover Multimode Microplate Reader (Promega), normalized with luminescence emitted by untreated cells, and elaborated with the GraphPad PRISM software version 6.01 (La Jolla, CA, USA) to calculate the half-maximal cytotoxic concentration (CC_50_). Based on the CC_50_, a non-toxic dose corresponding to 90–100% cell viability was used for each compound as the maximum concentration in the antiviral assays.

To determine the antiviral activity of the drugs, 4-fold decreasing concentrations of remdesivir, nirmatrelvir and EIDD-1931 were added to 5000 pre-seeded VERO E6 cells as described [11] and viral isolates were used at MOI 0.005 (corresponding to 100 TCID_50_) to infect the cultures after one hour. After 72 h incubation, antiviral drug activity was determined by measuring cell viability with the Cell Titer-Glo protocol as described above and expressed as half-maximal inhibitory drug concentration (IC_50_). Infected and uninfected cells without drugs were used to calculate the 100% and 0% of viral replication, respectively. Drugs were tested in technical duplicates in at least two independent experiments. The experiments were performed in the absence and in the presence of the P-gp inhibitor to evaluate the impact of the efflux system on the different compounds. The P-gp inhibitor concentration was set at 0.5 µM based on cytotoxicity data (Appendix A) and previous literature [11,12].

### 2.5. Statistical Analysis

Data were expressed as median followed by interquartile range [IQR] as appropriate for the distribution of data based on the Shapiro–Wilk test for normality. The Kruskal–Wallis test followed by Mann–Whitney test post hoc analysis was used to compare independent groups, whereas the Friedman test followed by Wilcoxon Rank Sum test post hoc analysis was used to compare multiple paired data. 

## 3. Results

### 3.1. mAb Treated Patients

Of 50 subjects screened, 40 were enrolled (19 males, 59.8 ± 17.7 years) including 6 who were previously vaccinated but lacked NtAb at baseline. Of these, 5 had haemato-oncologic disease and the remaining one was a 69-year-old male with underlying hypertension, chronic ischemic cardiomyopathy, dyslipidemia. Only one of the enrolled patients was asymptomatic whereas the others had mild symptoms of SARS-CoV-2 infection including fever (67.5%, *n* = 27), cough (65.0%, *n* = 26), headache (37.5%, *n* = 15), arthomyalgia (30.0%, *n* = 12), dysgeusia (17.5%, *n* = 7), gastrointestinal disorders (12.5%, *n* = 5), and dyspnea (5.0%, *n* = 2). Comorbidities and detailed information about enrolled individuals are indicated in Appendix A. Patients were treated with BAM/ETE (*n* = 12), CAS/IMD (*n* = 14), or SOT (*n* = 14) starting 3.7 ± 1.6 days from diagnosis.

Two patients were hospitalized (one in BAM/ETE and one in the CAS/IMD group), the others resolved SARS-CoV-2 infection without clinical complications. The time from mAb infusion to SARS-CoV-2 RNA negativization was available only in 22 individuals treated with BAM/ETE (*n* = 11) or CAS/IMD (*n* = 11) and was not significantly different with the two cocktails (15 vs. 13 days for BAM/ETE and CAS/IMD, respectively). 

### 3.2. Neutralizing Activity of mAbs against Different Viral Variants

In post-infusion sera, BAM/ETE, CAS/IMD, and SOT showed activity against the wild type (6295 (4355–8075) ID_50_ for BAM/ETE; 18,214 (16,248–21,365) ID_50_ for CAS/IMD; and 456 (265–592) ID_50_ for SOT) and the delta (14,780 (10,905–21,020) ID_50_ for BAM/ETE, 63,937 (47,211–79,971) ID_50_ for CAS/IMD, and 1103 (843–1334) ID_50_ for SOT). However, BA.1 was neutralized only by SOT (200 (37–233) ID_50_) whereas BA.2 was neutralized by CAS/IMD (174 (134–209) ID_50_) and SOT (20 (9–31) ID_50_), but not by BAM/ETE (Figure 1). 

When NtAb titers were analyzed as fold-change (FC) with respect to the wild-type strain, BAM/ETE, CAS/IMD and SOT neutralized the delta variant with 2.5 (1.8–3.6), 3.5 (2.4–5.1) and 2.1 (1.6–3.4) FC increase, respectively (all *p* < 0.001). With respect to wild type, SOT neutralizing activity decreased more with BA.2 (23.9 (14.2–43.5) FC) than with BA.1 (2.8 (1.1–4.1) FC) (*p* < 0.001). The partially regained activity of CAS/IMD against BA.2 was 99.3 (91.8–138.5) fold lower than that against the wild-type virus, a larger FC decrease compared with SOT (*p* < 0.001), although the absolute NtAb titer of CAS/IMD remained higher than that of SOT, due to the lower dosage and/or intrinsic activity of the latter. Indeed, NtAb titers were significantly higher for CAS/IMD vs. BAM/ETE and for both CAS/IMD and BAM/ETE vs. SOT against the wild-type and delta virus, as well as for CAS/IMD vs. SOT against BA.2 (*p* < 0.001 for all comparisons). 

### 3.3. Antiviral Activity of Nirmatrelvir, Molnupiravir and Remdesivir in VERO E6

Table 1 shows the cytotoxicity and the antiviral activity data for EIDD-1931, nirmatrelvir and remdesivir. No significant differences were observed for any drug IC_50_ across the viral variants considered. The impact of the P-gp inhibitor, as measured with the wild-type virus, was negligible with EIDD-1931 but highly relevant with remdesivir and nirmatrelvir, resulting in increased antiviral activity by 88- and 126-fold, respectively (Figure 2).

## 4. Discussion

Despite a limited evolutionary rate, continuous massive worldwide replication of SARS-CoV-2 has generated an array of mutants, with new variants typically outpacing past lineages and quickly becoming dominant [13]. Not surprisingly, most mutations in evolutionarily successful variants have occurred in the spike glycoprotein resulting in improved virus entry and increased transmissibility [14]. First detected in late 2021, the omicron variant led a major shift in SARS-CoV-2 evolution [15], driven by an unprecedented number of spike mutations and further evolving into a constellation of related lineages including BA.1, BA.1.1, BA.2 and later BA.3, BA.4 and BA.5, with some sublineages spreading faster than others in specific countries such BA.2.12.1 in the US [6]. A major consequence of omicron divergence from past lineages is the markedly reduced neutralization by sera from individuals recovering from natural infection with previously dominating variants and/or immunized with vaccines derived from the ancestral virus strain [16]. Likewise, several mAbs based on virus variants dominating the first epidemic waves have lost activity against omicron lineages [17].

Unlike the other licensed mAbs, SOT was derived from the antibody repertoire of an individual recovered from SARS-CoV in 2003 and shown to be cross-reactive to SARS-CoV-2, thus targeting a highly conserved domain [18]. Indeed, when compared with BAM/ETE and CAS/IMD, SOT had the smallest-fold decrease in activity against omicron BA.1 and BA.2 with respect to the ancestral reference virus, both in previous in vitro studies [19,20] and in this ex vivo study. However, we observed higher absolute NtAb titers to BA.2 with CAS/IMD compared to SOT in our ex vivo assay. This apparently contradictory result likely derived from the combination of three factors. First, IMD may have residual activity against BA.2, despite a fold decrease with respect to the ancestral virus ranging from 20 to 500 [4,19,20,21]. Second, the in vivo dosage of CAS/IMD is higher than that of SOT (1200 plus 1200 mg vs. 500 mg). Third, the intrinsic in vitro neutralizing activity of SOT is one order of magnitude lower than that of CAS or IMD, as indicated by EC_50_ values with the susceptible wild-type virus [3,22,23].

At present, it is unclear how this expected activity, for both SOT and CAS/IMD, can translate into clinical benefit with BA.2 infection. It must be emphasized that in vitro neutralization assays can capture just one component of the mAbs activity. Indeed, unlike other mAbs, neither SOT nor CAS/IMD have been engineered to remove effector functions such as engagement of Fc receptors, and SOT was recently shown to trigger antibody-dependent cytotoxicity and phagocytosis [5,24]. Of note, both SOT and CAS/IMD, as well as CIL/TIX, have been recently reported to curb experimental disease progression in the BA.2 infected hamster model, as shown by decreased infectious virus titer in the lungs by a factor which was comparable with the D614G infected control animals [25].

As opposed to mAbs variant-dependent activity, it was reassuring to confirm that the three licensed antivirals retain their full potency in vitro against the BA.1 and BA.2 omicron lineages. Of note, there has been only one report documenting this activity against the currently dominating BA.2 variant in vitro [4]. While VERO cells were used both in the previous and in our study, we extended the analysis by testing drug activity both in the presence and in the absence of the CP-100356 hydrochloride P-gp inhibitor. This is important because VERO cells overexpress P-gp, a condition which should not occur in human SARS-CoV-2 cell targets in vivo. In addition, nirmatrelvir is administered in vivo together with ritonavir which inhibits the P450 cytochrome CYP3A4 isoenzyme as well as P-gp [26]. Our results without the P-gp inhibitor closely matched the activity shown in the previous in vitro work under the same experimental conditions but we also showed that under P-gp inhibitor treatment the activity is enhanced around 100-fold for nirmatrelvir and for remdesivir. In addition, we comprehensively tested the cytotoxicity of the P-gp inhibitor in the same cell line, both alone and in combination with the different drugs, thus analyzing drug/P-gp inhibitor interactions. Indeed, the P-gp inhibitor decreased cell viability by 20% at 2 µM, a concentration which has been used in some of the previous studies evaluating the antivirals against past virus lineages [27,28]. This data helps to define how to measure antiviral activity of current and future antivirals.

Thus, our work strengthens the concept of resilience to SARS-CoV-2 variability with antivirals as opposed to the continuous challenge with mAbs. However, both antivirals and mAbs should be retested with any new virus variant. We believe our work contributes to the need to set technical aspects and procedures to comprehensively define the potential of the antiviral armamentarium, including mAbs and small molecules, and keep pace with virus variability during the ongoing pandemic.

## Figures and Tables

**Figure 1 viruses-14-01374-f001:**
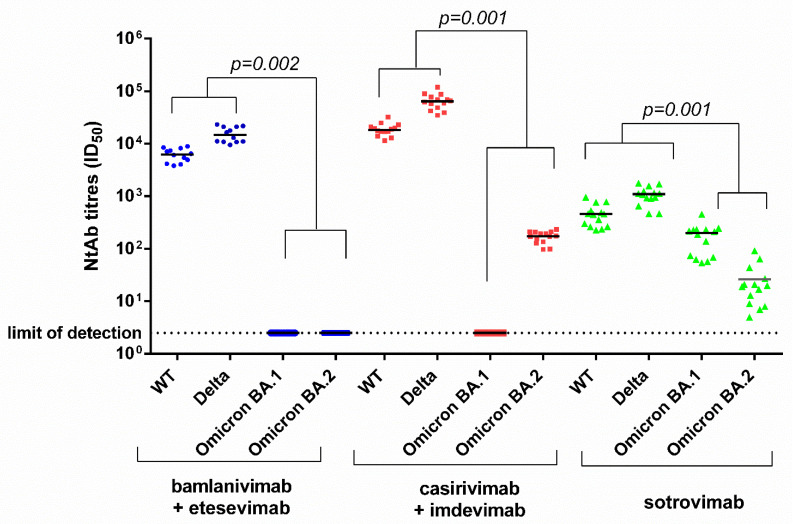
Ex vivo anti-SARS-CoV-2 wild type, delta, omicron (BA.1 and BA.2) neutralizing antibody titers measured in sera from 40 patients following infusion of bamlanivimab/etesevimab, casirivimab/imdevimab, or sotrovimab monoclonal antibodies. Blue dots, red squares and green triangles represents patients treated with bamlanivimab + etesevimab, casirivimab + indevimab and sotrovimab, respectively. Paired data were analyzed by the non-parametric Wilcoxon Signed Rank Sum test. NtAb titers before infusion were negative against each variant tested (not shown in figure). NtAb: neutralizing antibody; ID_50_: the reciprocal value of the sera dilution showing the 50% protection of virus-induced cytopathic effect; WT: wild type.

**Figure 2 viruses-14-01374-f002:**
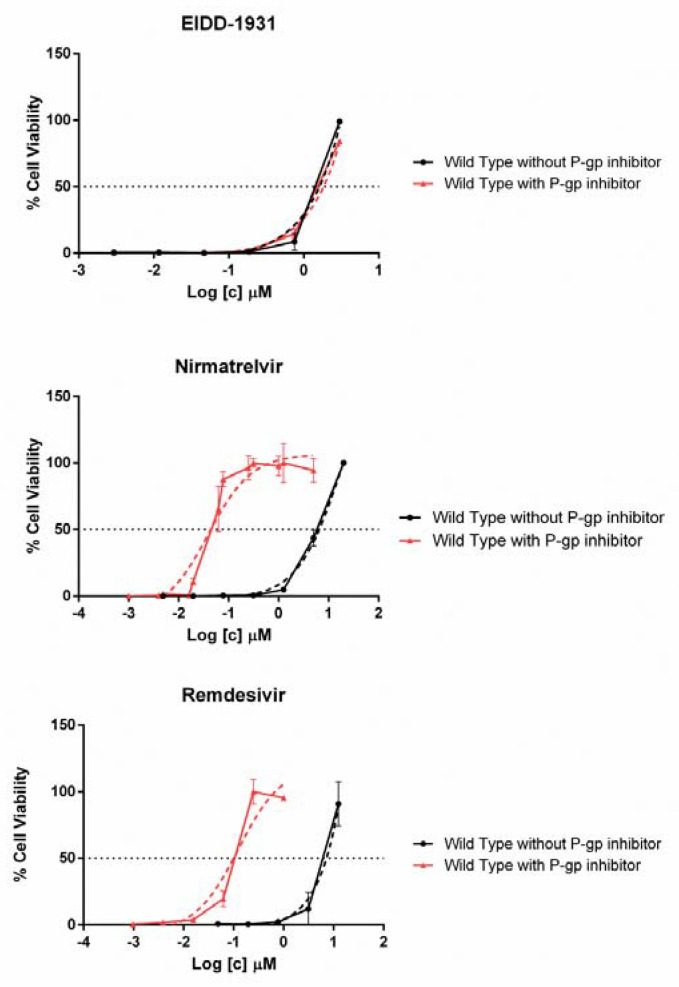
Comparison between antiviral activity of EIDD-1931, Nirmatrelvir and Remdesivir against wild-type SARS-CoV-2 virus with (0.5 µM) or without the addition of P-gp inhibitor. On the x-axis is indicated the micromolar drug concentration in logarithmic scale. The horizontal dashed line indicates the drug IC_50_ corresponding to 50% cell viability whereas the dashed curves indicate the dose response fitting curve generated by GraphPad PRISM software version 6.01 (La Jolla, CA, USA).

**Table 1 viruses-14-01374-t001:** Anti-SARS-CoV-2 activity of EIDD-1931 (the active form of molnupiravir), remdesivir and nirmatrelvir in VERO E6 cells. Compounds were tested in absence of P-gp inhibitor (CP-100356 hydrochloride) against wild-type strain and in presence of 0.5 µM P-gp inhibitor against Wild Type (WT), Delta, BA.1, and BA.2 variants. CC_50_: half-maximal toxic drug concentration; IC_50_: half-maximal inhibitor drug concentration; SD: Standard Deviation.

Compound	CC_50_ (µM)	IC_50_ WT (µM) Mean ± SD	IC_50_ Delta (µM) Mean ± SD	IC_50_ BA.1 (µM) Mean ± SD	IC_50_ BA.2 (µM) Mean ± SD
EIDD-1931	40.6 ± 3.7	1.10 ± 0.10			
EIDD-1931 plus P-gp inhibitor	43.3 ± 6.0	1.50 ± 0.10	1.50 ± 0.70	1.00 ± 0.50	0.80 ± 0.01
Nirmatrelvir	69.6 ± 1.0	5.80 ± 0.80			
Nirmatrelvir plus P-gp inhibitor	40.7 ± 4.4	0.05 ± 0.02	0.06 ± 0.01	0.04 ± 0.02	0.04 ± 0.01
Remdesivir	205.0 ± 35.4	6.90 ± 2.30			
Remdesivir plus P-gp inhibitor	17.2 ± 0.2	0.08 ± 0.04	0.11 ± 0.08	0.05 ± 0.04	0.08 ± 0.01

## Data Availability

Not applicable.

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
