# Peer review of "Efficacy of Licensed Monoclonal Antibodies and Antiviral Agents against the SARS-CoV-2 Omicron Sublineages BA.1 and BA.2"

_viruses, 2022, doi:10.3390/v14071374_

Round 1

Reviewer 1 Report

The authors investigated the antiviral activity of monoclonal antibodies that are/ have been used for the treatment of COVID19 patients against the omicron sublineages BA.1 and BA.2.

My major point of critisism is that part of this study have already been published by the authors. Further, there is no indication in the current manuscript that some data have already been published elsewhere. The supplemental table of the current manuscript and the previous manuscript (Dragoni et al, 2022, Clinical Microbiology and Infection) indicate that the same serum samples have been used. However, those serum samples have not been measured twice to obtain data presented in the current manuscript. Instead, already published date as shown in Dragoni et al. were included in the current figures without any comment on duplicate publication. E.g. current manuscript data on wildtype and SOT, line 171 and Fig. 1: ID50 = 456 [265-592] vs. Dragoni et al Fig 1: wildtype and sotrovimab =456 (259-592). Also some of the data obtained for Delta and omicron (BA.1) seem to be re-used in the current manuscript. Compared to Dragoni et al., the only novelity included in this manuscript are data on BA.2.

Minor Points:

1. The authors did not comment on the withdrawal of bamlanivimab.

2. Given the emergence of novel omicron sublineages, the panel of lineages analyzed should at least comprise BA.4 and BA.5.

3. Measuring cell viability as a readout for neutralization capacity of sera is an indirect method to determine viral replication compared to assays aiming to determine PFU or TCID50. Therefore, the latter should be used to evaluate antiviral activity of particular compounds/ sera.

4. Fig. 2: What do  the dashed lines indicate?

5. Fig. 2: Data on omicron sublineages should be included.

Author Response

Reply to major point 1. As observed by the reviewer, this study added newly obtained sera to those collected and analyzed in our previously published work (Dragoni et al., doi: 10.1016/j.cmi.2022.03.005). In detail, in Dragoni et al., we used 30 sera collected from 10 patients treated by the Eli-Lilly cocktail, 10 with the Regeneron cocktail and 10 with Sotrovimab. In current work, we increased the number of sera to 12 (El-Lilly), 14 (Regeneron) and 14 (Sotrovimab) to reach the final number of 40. Indeed, as noted by the reviewer, the median NtAb value of Sotrovimab sera vs. wild type virus was the same but with different Q3 (265 vs. 259). To ensure consistency with previous work, a random selection of 15 sera included in the previous work were retested vs. the wild type virus and found to yield comparable values (data not shown). Reasons for the sample overlap include: (i) the difficulty to find new cases treated by the Eli-Lilly and Regeneron cocktail since the administration of these mAbs was halted and, more generally, (ii) the difficult to meet the inclusion criterion of lack of SARS-CoV-2 NtAbs at baseline due to high rate of infection and/or vaccination in the whole population.

Of course, it was not our intention to hide that this work includes sera previously used in our previous work and indeed we cited that previous work (ref. 9 in the original submission, ref.8 in the revised version). However, we agree that this point must be clarified and thus we mention the previous at the end of the Introduction section when the aim of the current work is explained. In addition, the Methods section also indicates the inclusion of 30 sera from previous work.

Minor Points:

  1. The authors did not comment on the withdrawal of bamlanivimab.

Reply to minor point 1. At present, according to the updated COVID-19 treatment guidelines, the distribution of bamlanivimab/etesevimab (El-Lilly cocktail), casirivimab/imdevimab (Regeneron cocktail) and Sotrovimab has been paused. Only bebtelovimab and the cilgavimab/tixagevimab cocktail as preventive therapy are still recommended. The manuscript was updated with this information at line 54-59. 

Given the emergence of novel omicron sublineages, the panel of lineages analyzed should at least comprise BA.4 and BA.5.

Reply to minor point 2. The suggestion of the reviewer is correct: inclusion of BA.4 and BA.5 (and possibly other sublineages) would make the work more complete. However, we cannot perform this analysis for two reasons:

  • At present, we do not have these sublineages ready to be tested. Since we are using authentic virus neutralization, the lineages must be isolated, sequenced by NGS to assign the variant, propagated, and titrated and finally used in the neutralization assay. The time necessary to perform this analysis, once isolates are available, is at least four weeks, which would considerably delay the resubmission of the manuscript.
  • Several sera would not be available for testing BA.4 and BA.5, considering that we started from a 1:5 dilution and performed the analysis at least in duplicate against each variant.

  1. Measuring cell viability as a readout for neutralization capacity of sera is an indirect method to determine viral replication compared to assays aiming to determine PFU or TCID50. Therefore, the latter should be used to evaluate antiviral activity of particular compounds/ sera.

Reply to minor point 3. We agree that plaque reduction assay is the gold standard system to determine virus neutralization, but this method is not amenable to testing large numbers of sera. Measuring ATP levels is an accurate estimate of cell viability which, when normalized with respect to mock infected cells and infected cells without serum, ensures that virus neutralization is specifically measured. Of note, the assay is quantitative and automated avoiding the operator dependent inconsistency which is typical of CPE scoring. When setting up the assay, we obtained comparable virus titration values with two different methods (Vicenti et al., doi: 10.1016/j.ejmech.2021.113683; ref 9 in revised version): (i) by plaque assay (PFU/ml) and (ii) by evaluating virus induced death in 6 replicates by CellTiter followed by Reed and Muench analysis. Based on this, we perform routinely the viral titration by TCID50/ml through CellTiter and the plaque assay only when a new reference strain is included in the neutralization panel.To clarify these technical aspects, we have added a paragraph in the Materials and Methods section.

“Before performing the neutralization and phenotypic experiments, six replicates of 5-fold serial dilution of each viral stock were titrated in VERO E6 cells to determine the tissue culture infectious doses per milliliter (TCID50/ml), which is defined as the amount of virus required to infect 50% of replicate cell cultures as previously described (Vicenti et al., doi: 10.1016/j.ejmech.2021.113683; ref 9 in revised version). The cytopathic effect and consequently the TCID50 was determined by the CellTiter-Glo 2.0 Luminescent Cell Viability Assay (Promega) as described below. In addition, each viral stock was initially quantified by plaque assay as previously published (Vicenti et al., doi: 10.1016/j.ejmech.2021.113683, ref 9 in revised version)” 

  1. Fig. 2: What do the dashed lines indicate?

Reply to minor point 4. The horizontal dashed line indicates the drug IC50 corresponding to 50% of cell viability while the dashed curve is the dose response fitting curve generated by Graphpad. We added this information in the figure legend.

  1. Fig. 2: Data on omicron sublineages should be included.

Reply to minor point 5. We decided to show the effect of the P-gp inhibitor (with and without) only on the wild type variant. Adding all the data derived from the other variants resulted in a cluttered graph due to almost complete overlap of data. The drugs indeed retain the same IC50 against all variants tested (Table 1), thus showing the effect of the P-gp inhibitor with one reference variant is appropriate.

Reviewer 2 Report

The article is well designed and written. In the last paragraph of the discussion section, only several sentences are needed to summarize the purpose and consequences of the study. The some example sentences have been written in the abstract section, it can be written that monoclonal antibodies may be insufficient due to mutations and that the antivirals may be a little more effective in the treatment of SARS CoV-2 infection.

Author Response

The article is well designed and written. In the last paragraph of the discussion section, only several sentences are needed to summarize the purpose and consequences of the study. The some example sentences have been written in the abstract section, it can be written that monoclonal antibodies may be insufficient due to mutations and that the antivirals may be a little more effective in the treatment of SARS CoV-2 infection.

Reply. We thank the reviewer for her/his appreciation of our work. We have remodeled the last part of the Discussion section as suggested.

Reviewer 3 Report

In this study, the authors assessed the ex vivo inhibition of the B.1 wild type (WT), delta and omicron BA.1 and BA.2 lineages by post-infusion sera from 40 individuals treated with bamlanivimab/etesevimab (BAM/ETE), casirivimab/imdevimab (CAS/IMD) and sotrovimab (SOT) as well as the activity of remdesivir, nirmatrelvir and molnupiravir. Results from this study showed that continued evolution of SARS-CoV-2 requires updating the mAbs arsenal, however antivirals have so far remained unaffected. 

Several suggestions:

1.     In line 56, please add a reference after [for emergency use].

2.     Line 90, please add the full name for BSL.

3.     Line 98, is there [50%] before [Tissue Culture Infectious Dose]?

4.     Line 146, please explain [IQR], or add a reference after this paragraph in line 150.

5.     In Fig. 2, in each treatment (e.g., wild-type without P-GP, one color), there are two lines. Are they representing two different experiments?

Round 2

Reviewer 1 Report

The authors have addressed the reviewer´s points. Especially the usage of sera/data already published in a recent manuscript has been clarified.